# Effects of SiC Fibers and Laminated Structure on Mechanical Properties of Ti–Al Laminated Composites

**DOI:** 10.3390/ma14061323

**Published:** 2021-03-10

**Authors:** Chenyang Hou, Shouyin Zhang, Zhijian Ma, Baiping Lu, Zhenjun Wang

**Affiliations:** School of Aeronautic Manufacturing Engineering, Nanchang Hangkong University, Nanchang 330063, China; houchenyang2157@163.com (C.H.); mazhijian2020@163.com (Z.M.); lbp029@163.com (B.L.); wangzhj@nchu.edu.cn (Z.W.)

**Keywords:** SiC fiber, laminated structure, Ti–Al laminated composite, in situ tensile, mechanical properties

## Abstract

Ti/Ti–Al and SiC_f_-reinforced Ti/Ti–Al laminated composites were fabricated through vacuum hot-pressure using pure Ti foils, pure Al foils and SiC fibers as raw materials. The effects of SiC fiber and a laminated structure on the properties of Ti–Al laminated composites were studied. A novel method of fiber weaving was implemented to arrange the SiC fibers, which can guarantee the equal spacing of the fibers without introducing other elements. Results showed that with a higher exerted pressure, a more compact structure with fewer Kirkendall holes can be obtained in SiC_f_-reinforced Ti/Ti–Al laminated composites. The tensile strength along the longitudinal direction of fibers was about 400 ± 10 MPa, which was 60% higher compared with the fabricated Ti/Ti–Al laminated composites with the same volume fraction (60%) of the Ti layer. An in situ tensile test was adopted to observe the deformation behavior and fracture mechanisms of the SiC_f_-reinforced Ti/Ti–Al laminated composites. Results showed that microcracks first occurred in the Ti–Al intermetallic layer.

## 1. Introduction

Due to presenting several advantages such as low density, high modulus of elasticity, good high-temperature creep strength and oxidation resistance, titanium aluminide (TiAl) based alloys have great potential in aerospace applications [1,2,3]. However, they suffer from a major challenge of low room temperature ductility. This was considered to be a significant barrier for these classes of alloys for use in structural components.

Inspired by the structural biological materials such as the abalone shell, animal bones and mammal teeth, etc. [4,5], Ti intermetallic multilayered composites [6,7,8,9,10], which possess improved toughness, fracture resistance and excellent creep resistance and have been extensively researched. By micro-, meso- and macrostructure designing and tailoring, specific functionality of Ti intermetallic multilayered composites can be achieved. The superior specific properties of this class of composites make them extremely attractive for high-performance aerospace applications [11]. Meanwhile, continuous SiC fibers with outstanding properties of high strength, high modulus and low density have successfully been introduced into titanium matrix material through the vacuum high-temperature pressing (VHP) method [4,12]. It has been found that SiC fiber-reinforced Ti intermetallic multilayered composites possess excellent toughness and fracture resistance.

An SiC_f_-reinforced Ti intermetallic multilayers composite was fabricated by Yu et al. [13]. Results showed that along the longitudinal direction of SiC fibers, the ultimate tensile and flexural strength and fracture toughness of the composite increased by 53%, 74% and 75%, respectively, while the elongation remained almost similar to the Ti intermetallic multilayer composite. Zhu et al. [4] fabricated a Ti intermetallic multilayered/SiC_f_-reinforced Ti matrix composite and found that with the introduction of SiC fibers, the tensile and the flexural strength of the hybrid composite along the longitudinal direction of the fibers increased by 57% and 92% compared with the composite without SiC fiber. Wang et al. [14] investigated the fatigue behavior and damage modeling of SCS-6/titanium/titanium aluminide hybrid laminated composite. Compared with SCS-6/Ti-6-4 composite, the SCS-6/Ti-6-4/Ti-25-10 hybrid laminated composite possessed improved fatigue behavior. Zhang et al. [15] also studied the mechanical behaviors and failure mechanisms of SiC_f_-reinforced Ti/Ti_2_AlNb laminated composite. The researches above show that SiC_f_-reinforced Ti intermetallic laminated composites provide an effective approach for tailoring mechanical propertied. This can be attributed to the crack deflection ability of the Ti intermetallic multilayered structure, as well as the reinforcement effect of continuous SiC fibers.

It was found that the majority of studies focused on strengthening ductile Ti layers with SiC fibers, followed by hybridization with intermetallic layers. This is principally because the ductile layer can serve as the compliant layer to suppress the initiation and propagation of residual stress placed on cracks near the interface [14]. Lin et al. [16] prepared an SiC_f_-Ti/Al_3_Ti laminated composite by using the vacuum hot-pressing sintering method and found that SiC fibers can provide strengthening and toughening effects for the brittle intermetallic and the laminated composites. The strengthening mechanism of the SiC_f_-reinforced intermetallic layer still needs to be clarified. Meanwhile, reaction products, thickness, bonding strength of the interface that between SiC fiber and its surrounding intermetallic matrix, as well as the fiber damage, play critical roles in the mechanical properties of the SiC_f_-reinforced Ti/TiAl laminated composite. The fiber arrangement and the enhancement mechanism need to be further investigated.

Hence, an SiC_f_-reinforced Ti/Ti–Al laminated composite was designed and fabricated with commercial pure Ti foils, Al foils and SiC fibers by the VHP method. The interface between the SiC fiber and the intermetallic matrix was investigated. Tensile tessst and three-point bending test were adopted to evaluate the mechanical properties of the composites. The deformation behavior and fracture mechanisms of the SiC_f_-reinforced Ti/Ti–Al laminated composite were studied by applying an in situ tensile test. Furthermore, the Ti/Ti–Al laminated composite was fabricated and investigated for comparison.

## 2. Experimental Procedures

### 2.1. Structure Design

Commercial pure Ti foils (200 μm thick), pure Al foils (20, 100, 200 μm thick) were cut into round foils with diameters of 100 mm. Table 1 shows the chemical compositions of the selected materials. SiC fibers fabricated by Beijing Institute of Aeronautical Materials (China) were chosen as the reinforcement component. These fibers have diameters of 120 μm with a tungsten core (16 μm in diameter) and a layer of β-SiC (50 μm thick), as shown in Figure 1. In addition, a layer of carbon (2 μm) was coated by chemical vapor deposition (CVD) to inhibit the reaction. The Ti foils were cleaned in aqueous HF solution (10 vol.%) while the Al foils were etched in aqueous NaOH solution (10 wt.%) for 2 min. Subsequently, the materials were rinsed with alcohol and processed with ultrasonic cleaning in distiller water, then were dried immediately for further preparation.

In order to avoid fiber aggregation in the prepared SiC_f_-reinforced Ti/Ti–Al laminated composite, a novel method of fiber weaving was implemented. In this method, equally spaced predrilled holes were drilled into the Al foils (with thickness 20 μm) by a steel needle with Φ200 μm. Then the fibers travelled up and down the foil material through the drilled holes (Figure 2a). In this work, the spacing is 5 mm in the X direction and 20 mm in the Y direction. Fiber braids with 5 mm equal spacing were prepared as shown in Figure 2b.

The metallic foils and SiC fiber braids were stacked according to the schematic illustration in Figure 3. Three laminated structures were used: Ti–Al–Ti (Figure 3a), Ti–Al–SiC_f_-Al–Ti (Figure 3b Pattern A) and Ti–SiC_f_–Al–SiC_f_-Ti (Figure 3b Pattern B). The volume fraction of Ti layers was about 55% as designed in the preform. Among them, the Ti/Ti–Al laminated composite used the “Ti–Al–Ti” laminated structures, while the SiC_f_-reinforced Ti/Ti–Al laminated composite was made up of Pattern A and Pattern B.

### 2.2. Sintering Process

The prepared assembly was placed in a graphite mold and was subsequently moved into a vacuum hot-press furnace for sintering. The preparation process includes three steps, as shown in Figure 4.

Firstly, the temperature was raised to 600 °C at a rate of 10 °C/min and held for 60 min under a pressure of 5 MPa to achieve a primary combination between the Ti and Al foils. Secondly, the temperature increased to 660 °C (the melting temperature of pure aluminum) for 2 h to ensure that Al foils were consumed completely, while the pressure was decreased to 0 MPa to avoid the expulsion of the molten aluminum. Finally, the temperature and pressure increased to 950 °C and 10 MPa for one hour for the Ti/Ti–Al laminated composite, or 950 °C and 40 MPa for one hour for the SiC_f_-reinforced Ti/Ti–Al laminated composite. Then, the exerted pressure was released, and the temperature decreased to room temperature in the furnace. Schematic illustration of the sintering parameters is presented in Figure 4.

### 2.3. Materials Characterization

After VHP, specimens used for characterization were cut from the synthesized samples with a wire electrical discharge machine (DK7355, Xiongfeng Machinery Co., Ltd., Ningbo, China), and were inlaid into epoxy resin. Subsequently, the specimens were ground using sandpaper and polished to a smooth mirror surface using diamond paste. Then, metallographic specimens were etched by reagent (5 vol.% HF+15 vol.% HNO_3_ + 80 vol.% H_2_O).

Optical microscopy (OM, IM 300, China) was performed for the microstructure observation. A Field Emission Scanning Electron Microscope (SEM; FEI Nova Nano SEM450, Hillsboro, OR, USA) equipped with an Energy Dispersive X-ray Spectrometer (EDXS, INCA 250X-Max 50, Oxford, UK) was used for the microstructure observation and local composition analysis. X-ray diffraction (XRD, D8 ADVANCE, Brooke, Germany) was performed for the phase identification.

### 2.4. Mechanical Properties Measurements

Quasi-static tensile tests were carried out on the fabricated Ti/Ti–Al and SiC_f_-reinforced Ti/Ti–Al laminated composites at room temperature with a loading rate of 0.2 mm/min. Test specimens were machined along the SiC fibers direction using wire electrical discharge. Subsequently, their surfaces were polished. The gauge sections were 26 mm in length, 6 mm in width and 3 mm in thickness, as illustrated in Figure 5a. The strain sheet was affixed to both sides of the specimen, and the strain value was recorded by the strain gauge.

For the three-point bending test, the constant loading rate was 2 mm/min. The specimens were 80 mm in length, 10 mm in width and 3 mm, as shown in Figure 5b. The specimens of SiCf-reinforced Ti/Ti–Al laminated composites with two different loading modes are shown in Figure 5c. The supporting distance is 40 mm. After tests, the fracture surfaces of tensile specimens were observed by SEM.

In situ tensile testing of specimen in SEM (MINI-MTS2000, Qiyue Technology Co., Ltd., Hangzhou, Zhejiang Province, China) was applied with a loading rate of 5 μm/s. In order to better observe the fracture and crack extension behavior of the SiC_f_-reinforced Ti/Ti–Al laminated composites, each surface was polished, especially the observation surface—i.e., the cross section of the specimen that is 10 mm in gauge length (Figure 6).

## 3. Results and Discussion

### 3.1. Microstructure Characterization

Figure 7a,b present the transverse section of the fabricated SiC_f_-reinforced Ti/Ti–Al laminated composite. The layers with dark colors are the titanium, while the layers with light color are the formed intermetallic, among which are the scattered SiC fibers (black dots). As described above, two different arrangements were designed in the preform: Ti–Al–SiC_f_–Al-Ti and Ti–SiC_f_–Al-SiC_f_–Ti. Figure 7b shows the structure illustration of the fabricated laminated composite. It can be found that the SiC fibers in Pattern A are almost at equal distances (5.0 ± 0.5 mm) along the laminate direction. This means that the fiber intervals can be guaranteed by using the fiber braid, as shown in Figure 2, while in Pattern B, two layers of fiber braid were placed between the Ti foils (Ti–SiC_f_–Al–SiC_f_–Ti). With the melt and reaction of Al foils, two layers of fiber braids were pressed together, resulting in SiC fiber aggregation.

Both the actual volume fractions of the Ti layer in the Ti/Ti–Al and SiC_f_-reinforced Ti/Ti–Al laminated composites were approximately 60%, as measured. This is 5% higher than the theoretical volume fraction, which was mainly due to the reaction and mutual diffusion of aluminum and titanium atoms at high temperatures.

Previous studies [17,18] have demonstrated that due to the different diffusion coefficient of Ti and Al atoms, Kirkendall holes will emerge when fabricating Ti/Ti–Al laminated composites. Figure 8a,b present microstructures of the Ti/Ti–Al laminated composite, while Figure 8c–f show the microstructures of the SiC_f_-reinforced Ti/Ti–Al laminated composite. In Al/Ti diffusion couple, during the net movement of atoms from Al to Ti caused by the different diffusion coefficient, vacancies will form and diffuse from the Ti toward Al layer [17,18]. It can be observed that obvious Kirkendall holes existed in the Ti/Ti–Al laminated composite (Figure 8b). By comparison, Kirkendall holes in the SiC_f_-reinforced Ti/Ti–Al laminated composites were much lower in number (Figure 8d). This may be attributed to the larger exerted pressure (40 MPa) during the fabricating processes, compared to the 10 MPa pressure applied in the Ti/Ti–Al laminated composites.

Figure 8e,f show the microstructures of SiC_f_-reinforced Ti/Ti–Al laminated composites corresponding to Pattern A and Pattern B, as depicted in Figure 3b. SiC fiber aggregation occurred on the Pattern B side. With small spacing between the SiC fibers, more cracks emerged (Figure 8f). This means that the “Ti–Al–SiC_f_–Al–Ti” laminated structure (Pattern A) should be preferred to obtain an equal spacing of the fiber arrangement.

Figure 9 shows the X-ray diffraction analysis of the laminated composites. The results suggest that the intermetallic phases contain Ti_3_Al, TiAl, TiAl_2_ and TiAl_3_. Additionally, Al phases were not found in XRD patterns, which indicates that Al was completely consumed in the sintering process.

Figure 10a,b depict the phase composition, phase distribution and interfacial evolution of the Ti/Ti–Al interface zone. Typically, EDXS line scans were made along the yellow line to investigate the chemical compositions of different layers. Afterward, EDXS point scanning was used to determine the chemical compositions of the four marked points depicted in Figure 10a,b. The analysis results are shown in Table 2. These reveal that the corresponding phases are Ti, Ti_3_Al, TiAl_2_ and TiAl_3_, respectively. The results are consistent with previous studies [4,19].

It can also observe that the concentrations of the elements Ti and Al present significant gradient changes in the SiC_f_-reinforced Ti/Ti–Al laminated composites (Figure 10b) in contrast with the Ti/Ti–Al laminated composites (Figure 10a). A conclusion can be made that a more stable TiAl_3_ phase can be obtained with an exerted pressure 40 MPa in the sintering process.

Figure 10c presents the interface of the SiC fiber and intermetallic matrix. An intact C coating can be found on the SiC fiber surface, which prevents the fiber damage in the laminated composites. Meanwhile, a clear thin gray layer (about 1 μm thick) was formed on the Ti–Al intermetallic side (Figure 10c). According to the results of the line scanning in Figure 10c and the point scanning in Table 3 (Spot 12), the reaction layer can be identified as TiC and Al_4_C_3_, as demonstrated in previous reports [4,12,20]. However, they were not detected in XRD (Figure 9) due to the low contents.

Figure 10c shows holes with short strips around SiC fibers and Figure 10b shows that similar morphologies exist in Ti–Al intermetallic compound layer. This is due to the Ti and Al atomic diffusion coefficient differences, resulting in the formation of Kirkendall holes.

### 3.2. Mechanical Properties

Table 4 shows the tensile strength and flexural strength of the fabricated laminated composites. Compared with Ti/Ti–Al laminated composites, the tensile strength of SiC_f_-reinforced Ti/Ti–Al laminated composite increases by 60%, while there is no significant increase in flexural strength. The three-point bending test of SiC_f_-reinforced Ti/Ti–Al laminated composite shows that the bending strength on the Pattern B side is 40 MPa more than that on the Pattern A side. This is due to the fact that the volume fraction of SiC fiber in Pattern B is twice as much as the Pattern A side, despite there being fiber aggregation on the Pattern B side.

Figure 11 presents the stress–strain curves of Ti/Ti–Al and SiC_f_-reinforced Ti/Ti–Al laminated composites in room temperature uniaxial tension tests. The yielding strength at 0.2% deformation of the Ti/Ti–Al laminated composite is 217 MPa and the elastic modulus is 61.17 GPa. The yielding strength at 0.2% deformation and the elastic modulus of the SiC_f_-reinforced Ti/Ti–Al laminated composite are 339 MPa and 101.05 GPa, respectively. Comparing to the Ti/Ti–Al laminated composite (with the same volume fraction, 60%), the yielding strength at 0.2% deformation of SiC_f_-reinforced Ti/Ti–Al laminated composite increases by 122 MPa, while the elastic module increases by 65%. This means that by introducing the SiC fibers, both the tensile strength and the resistance to deformation of laminated composites are significantly improved. Meanwhile, the ultimate elongation of SiC_f_-reinforced Ti/Ti–Al laminate composites reaches 1.6%, which is about 14% higher than the Ti/Ti–Al laminate composites.

Figure 12 shows the force–displacement curves of SiC_f_-reinforced Ti/Ti–Al laminated composite obtained through in situ tensile testing. The curve consists of five stages: elastic deformation stage (I), yield stage (II), delamination stage (III), start failure stage (IV) and fracture stage (V). The entire failure process of in situ tensile testing was observed under SEM. Figure 13 describes the detailed deformation morphologies corresponding to different tension force values. Firstly, microcracks initiated in the Ti–Al intermetallic layer when the tensile load ranged from 1002 N to 1102 N, as shown in Figure 13a. As the load continued to increase, the microcracks propagated and merged in the Ti–Al intermetallic layers. Interlayer cracks appeared when the load reached 1109 N. The microcracks and interlayer cracks first appeared in the outer layer of the laminated composite, as shown in Figure 13a,b. When the load reached 1124 N, delamination occurred within the Ti–Al intermetallic layers (Figure 13c). Then the SiCf-reinforced Ti/Ti–Al laminated composite began to fail (Figure 13d). It can be observed that the outer layers failed first and then the inner layers (Figure 13e). Ultimately, the whole specimen fractured completely (Figure 13f).

The fractures of the fabricated laminated composites are composed of the ductile fracture (Ti layer) and the brittle fracture (Ti–Al intermetallic layer). Figure 14 and Figure 15 show the tensile fracture morphologies of Ti/Ti–Al and SiC_f_-reinforced Ti/Ti–Al laminate composites. It can be seen that delamination was mainly generated in the Ti–Al intermetallic layer of the prepared laminated composites (Figure 14a and Figure 15a). Both trans-granular and intergranular fractures were observed (Figure 14c). Additionally, secondary cracks (Figure 14b) emerged in the brittle Ti–Al intermetallic layer, while the Ti layer closely combined with the Ti–Al intermetallic layer, and there is no delamination in the Ti layer (Figure 14e and Figure 15e).

Fiber reinforcement consists of fiber pullout, fiber debonding and fiber fracture. In this experiment, it can be seen that fiber fracture was the main failure mode. Debonding of SiC fibers can also be observed at the interface between the fracture SiC fiber and the intermetallic matrix (Figure 15b). No pulling out failure mode was found at the fracture section, which can be ascribed to the high interface bonding strength. The excessive interface binding strength of the fibers with the matrix may limit the enhancement effect of fibers to some extent, consistent with the results in the literature [21,22]. There is no damage of the fibers in the tensile experiment (Figure 15c,f). Meanwhile, the interface between the Ti layer and the Ti–Al intermetallic layer was well integrated, which further verified that with the higher exerted pressure, more compact structures with fewer Kirkendall holes can be obtained. The conclusion can be drawn that with the introduction of the fibers, the fracture resistance of the SiC_f_-reinforced Ti/Ti–Al laminated composites further improves.

Figure 16 presents the load–deflection curves for the three-point bending test of Ti/Ti–Al and SiC_f_-reinforced Ti/Ti–Al laminated composites. According to the results, load–deflection curves can be divided into four stages. At the beginning of loading (Stage I), the curves presented a linear increase. The flexural stiffness at this stage ranges from 87 to ~104 GPa, as calculated. When the value of load reaches about 350 N (Stage II), the prepared laminated composites start to fail. The flexural strengths of Ti/Ti–Al and SiC_f_-reinforced Ti/Ti–Al laminated composites are close to 900~950 MPa. It can be observed that the bending performance of laminated composites was only slightly improved due to the small amount of unidirectional SiC fibers introduced.

Figure 17 shows the OM images of the bending crack morphologies. During the three-point bending test, delamination occurred on the tensile stress side of the prepared laminated composites. The fracture of the outermost layer played a critical role in determining the bending failure. The fracture and stripping of fibers can be observed in Figure 17d,f.

Figure 18 shows the bending crack morphologies of the fabricated laminated composites. Firstly, the microcracks in the Ti–Al intermetallic layers initiated parallel to the loading direction (Figure 18b,e,g). Subsequently, the microcracks continued to spread to the Ti layer. Because of the obstruction of the ductile titanium layer, the crack tip blunted and did not continued to spread in the Ti layer (Stage I, Figure 18c,e,g). With the increase in load (Stage II), more microcracks were generated. With the microcracks’ growth and merging, long cracks parallel to the laminate structure emerged. These cracks propagating in the Ti–Al intermetallic layers led to the delamination of laminated composites. The propagation path of the cracks was long and zigzagged (Stage III, Figure 18a,d,f), which indicates a better absorption of fracture energy.

## 4. Conclusions

Ti/Ti–Al and SiC_f_-reinforced Ti/Ti–Al laminated composites were fabricated through vacuum hot-pressure, and the effects of SiC fiber and laminated structure on the properties of Ti–Al laminated composites were studied. The main conclusions are presented as follows:Equal spacing of the fibers could be guaranteed for SiC_f_-reinforced Ti/Ti–Al laminated composites prepared by a novel method of fiber weaving. No other elements would be introduced to contaminate the composites.With the higher exerted pressure, more compact structure with fewer Kirkendall holes could be obtained in SiC_f_-reinforced Ti/Ti–Al laminated composites.SiC_f_-reinforced Ti/Ti–Al laminated composites had a tensile strength of 400 ± 10 MPa and flexural strength of 900 ~ 950 MPa. Compared to Ti/Ti–Al laminates, the tensile strength increased by 60%, while the ultimate elongation reached 1.6% (increased by about 14%). The flexural strength did not change much (Ti/Ti–Al laminate composites had a flexural strength of 923 ± 10 MPa). The tensile properties of the laminated composites could be effectively improved by introducing the SiC fibers, while the bending properties were not obviously influenced due to the small volume fraction of fibers.The deformation behavior and fracture mechanisms of SiC_f_-reinforced Ti/Ti–Al laminated composites were obtained through in situ tensile tests. Microcracks first occurred in the Ti–Al intermetallic layer. With the growth and merging of microcracks, interlayer cracks formed in the Ti–Al intermetallic layer along the load direction.

## Figures and Tables

**Figure 1 materials-14-01323-f001:**
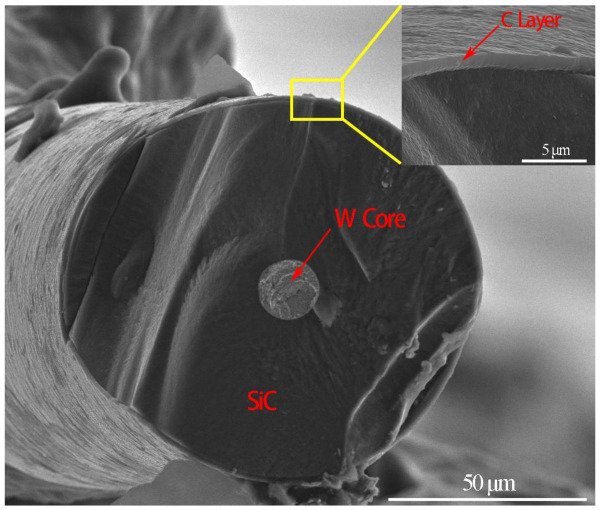
Morphology of SiC fiber.

**Figure 2 materials-14-01323-f002:**
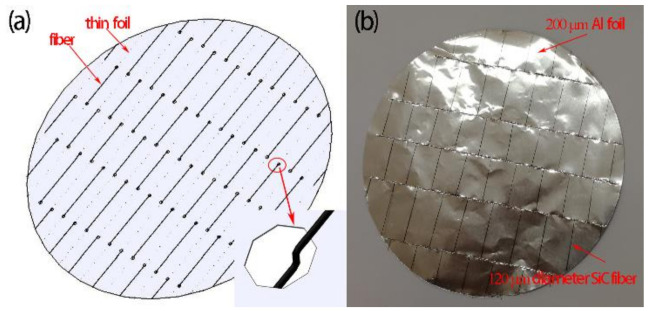
Fiber braid: (**a**) schematic illustration; (**b**) fiber braid.

**Figure 3 materials-14-01323-f003:**
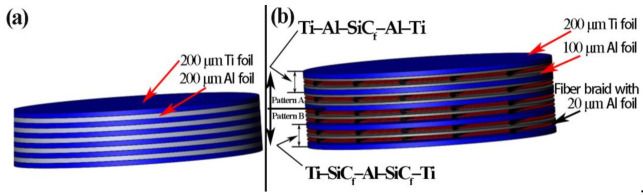
Schematic illustration of the stacking sequence: (**a**) Ti–Al–Ti; (**b**) Pattern A: Ti–Al–SiC_f_–Al–Ti and Pattern B: Ti–SiC_f_–Al–SiC_f_–Ti.

**Figure 4 materials-14-01323-f004:**
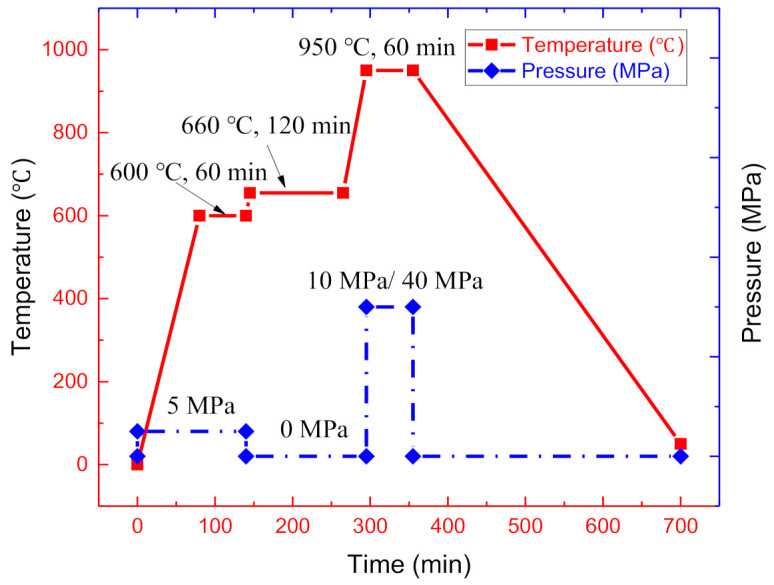
Schematic illustration of the sintering parameters.

**Figure 5 materials-14-01323-f005:**
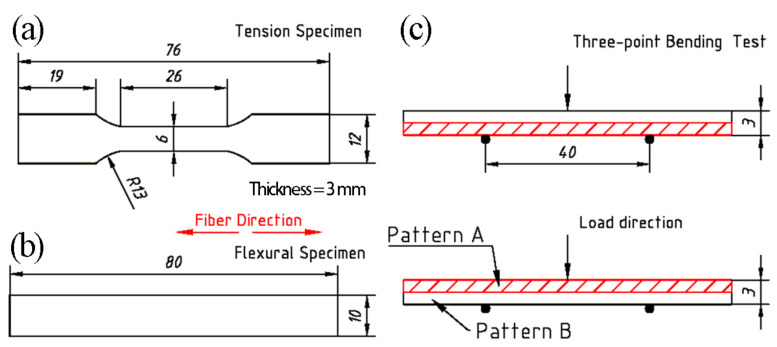
Schematic illustrations of the specimens (units: mm): (**a**) tensile specimen; (**b**) flexural specimen; (**c**) three-point bending test (Pattern A: Ti–Al-SiC_f_-Al-Ti; Pattern B: Ti-SiC_f_-Al-SiC_f_-Ti).

**Figure 6 materials-14-01323-f006:**
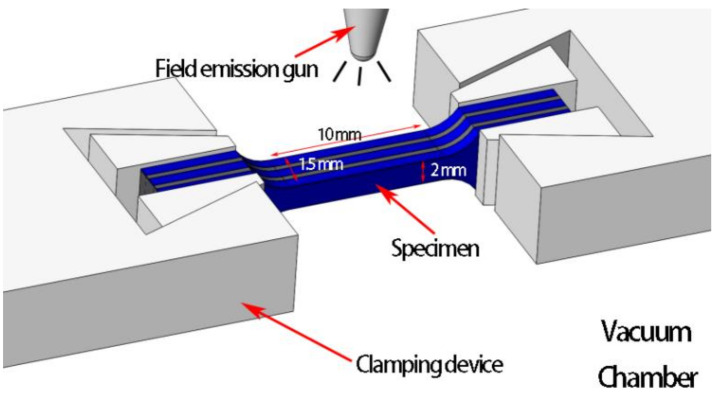
Schematic illustrations of the in-situ tension specimen (unit: mm).

**Figure 7 materials-14-01323-f007:**
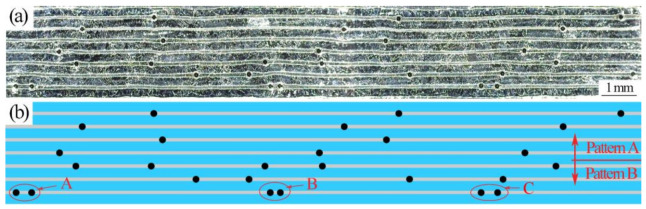
Transverse section of SiC_f_-reinforced Ti/Ti–Al laminated composites: (**a**) Optical microscopy (OM) image; (**b**) structure illustration.

**Figure 8 materials-14-01323-f008:**
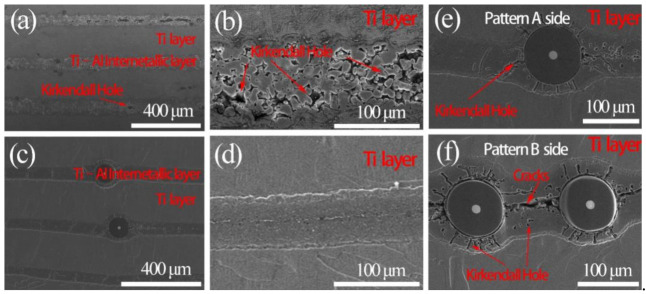
Microstructure of the fabricated laminated composites: (**a**), (**b**) Ti/Ti–Al laminated composite; (**c**–**f**) SiC_f_-reinforced Ti/Ti–Al laminated composites.

**Figure 9 materials-14-01323-f009:**
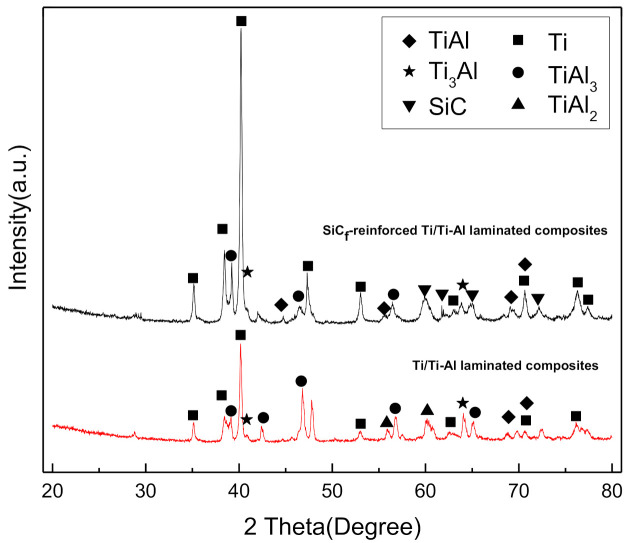
X-ray diffraction (XRD) patterns of the fabricated Ti/Ti–Al laminated composites.

**Figure 10 materials-14-01323-f010:**
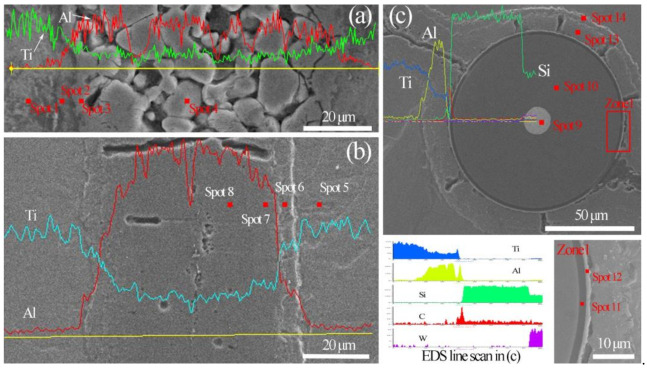
EDXS results: (**a**) Ti/Ti–Al laminated composites, (**b**) SiC_f_-reinforced Ti/Ti–Al laminated composites, and (**c**) SiC_f_/Ti–Al interface.

**Figure 11 materials-14-01323-f011:**
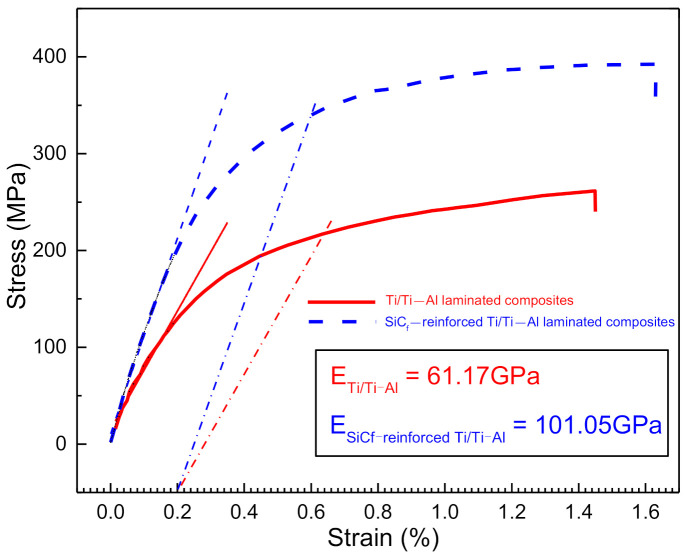
Stress–strain curves of the fabricated laminated composites.

**Figure 12 materials-14-01323-f012:**
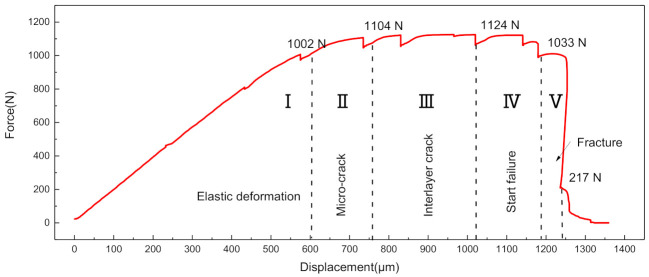
Force–displacement curve of SiC_f_-reinforced Ti/Ti–Al laminate composite under in situ tensile testing.

**Figure 13 materials-14-01323-f013:**
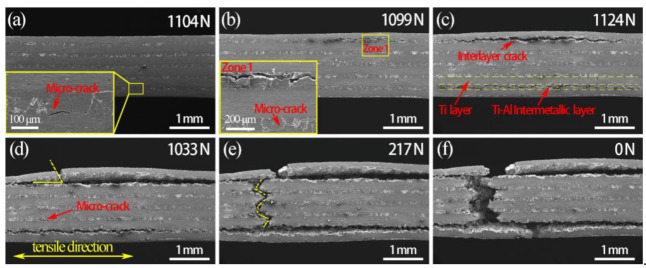
In situ SEM micrograph at the tensile force of (**a**) 1104 N, (**b**) 1119 N, (**c**) 1124 N, (**d**) 1033 N, (**e**) 217 N and (**f**) 0 N.

**Figure 14 materials-14-01323-f014:**
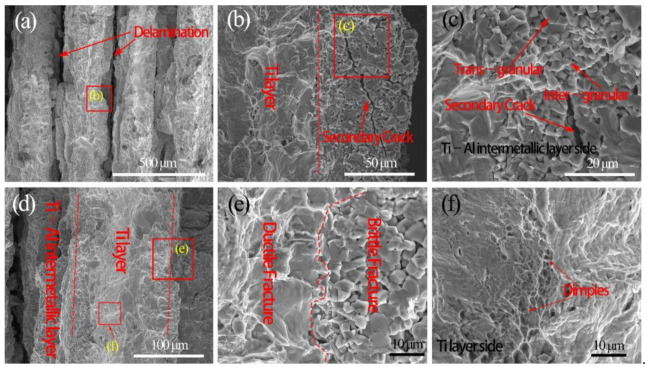
The tensile fracture morphologies of Ti/Ti–Al laminated composite: (**a**) Delamination in the Ti–Al intermetallic layer, (**b**) Secondary Crack in the Ti–Al intermetallic layer, (**c**) Trans–granular and Inter–granular in the Ti–Al intermetallic layer side, (**d**) Interface between Ti–Al intermetallic layer and Ti layer, (**e**) Interface between Ductile Fracture and Brittle Fracture, (**f**) Dimples in the Ti layer side.

**Figure 15 materials-14-01323-f015:**
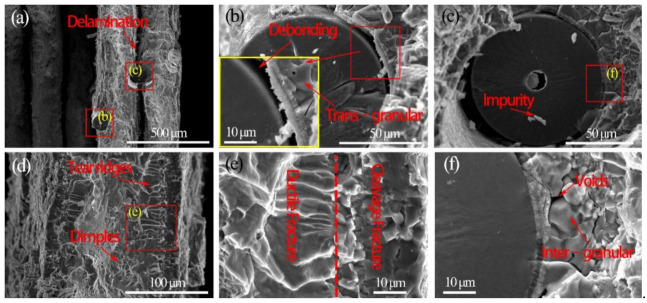
The tensile fracture surfaces of SiC_f_-reinforced Ti/Ti–Al laminated composite: (**a**) Delamination in the Ti–Al intermetallic layer, (**b**) Debonding between fracture SiC fiber and the intermetallic matrix, (**c**) Interface between SiC fiber and the intermetallic matrix, (**d**) Dimples and Tear ridges in the Ti layer, (**e**) Interface between Ductile Fracture and Brittle Fracture, (**f**) Inter–granular in the Ti–Al intermetallic layer.

**Figure 16 materials-14-01323-f016:**
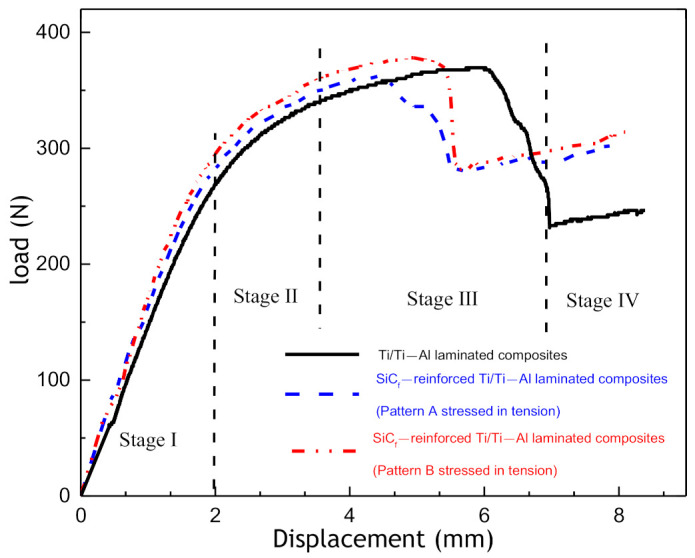
Load–displacement curves of the fabricated laminated composites.

**Figure 17 materials-14-01323-f017:**
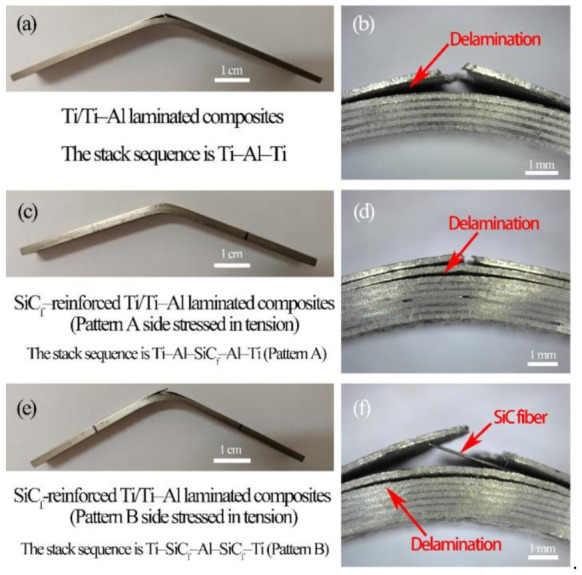
OM image of crack morphologies in three-point bending test: (**a**) and (**b**) Ti/Ti–Al laminated composites, (**c**) and (**d**) SiC_f_-reinforced Ti/Ti–Al laminated composites (Pattern A side stressed in tension), (**e**) and (**f**) SiC_f_-reinforced Ti/Ti–Al laminated composites (Pattern B side stressed in tension).

**Figure 18 materials-14-01323-f018:**
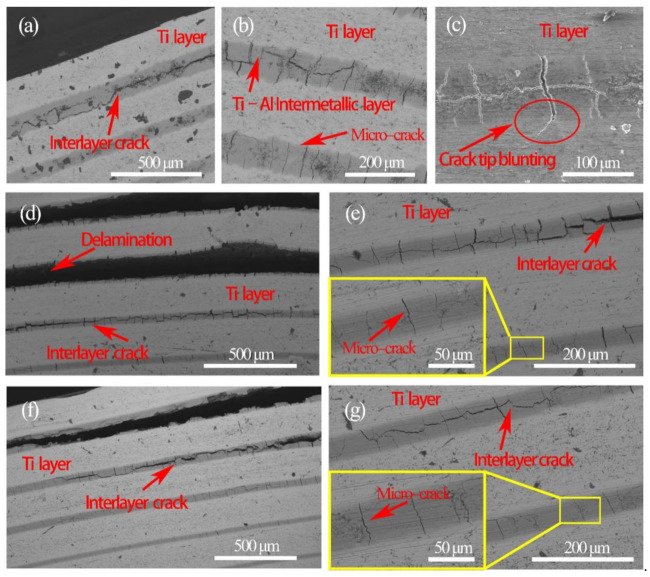
The crack morphologies of the prepared laminated composites in three-point bending test: (**a**–**c**) Ti/ Ti–Al Laminated Composite; (**d**–**g**) SiC_f_-reinforced Ti/ Ti–Al laminated composites, (**d**) and (**e**) Pattern A stressed in tension, (**f**) and (**g**) Pattern B stressed in tension.

**Table 1 materials-14-01323-t001:** Chemical compositions of TA1 titanium and 1060 aluminum alloy (wt.%).

Materials	Composition (%)
TA1	Ti: Margin Fe < 0.13, C < 0.012, N < 0.003, O < 0.11, H < 0.003
1060Al	Al: Margin Si0.25, Fe0.35, Cu0.05, Mn0.3, Mg0.03, Zn0.05, Ti0.03

**Table 2 materials-14-01323-t002:** Typical chemical compositions of the Ti/Ti–Al interface zone detected by Energy Dispersive X-ray Spectrometry (EDXS).

Point No.	Ti	Al	Point No.	Ti	Al	Phase
Spot 1	95	5	Spot 5	96	4	Ti
Spot 2	70	30	Spot 6	71	29	Ti_3_Al
Spot 3	35	65	Spot 7	34	66	TiAl_2_
Spot 4	24	76	Spot 8	25	75	TiAl_3_

**Table 3 materials-14-01323-t003:** Typical chemical compositions of the SiC_f_/Ti–Al interface area detected by EDXS.

Point No.	Ti	Al	Si	C	W
Spot 9	–	–	–	–	100
Spot 10	–	–	36	64	–
Spot 11	0	0	5	95	–
Spot 12	28	8	3	61	–
Spot 13	34	66	–	–	–
Spot 14	72	28	–	–	–

**Table 4 materials-14-01323-t004:** Mechanical performance results of Ti/Ti–Al and SiC_f_-reinforced Ti/Ti–Al laminated composites.

Composites	Tensile Strength (MPa)	Flexural Strength (MPa)
SiC_f_-reinforced Ti/Ti–Al laminated composites	400 ± 10	910 ± 30 Pattern A stressed in tension
		950 ± 30 Pattern B stressed in tension
Ti/Ti–Al laminated composites	250 ± 30	923 ± 10

## Data Availability

Data sharing not available.

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
