# Peer review of "Effects of SiC Fibers and Laminated Structure on Mechanical Properties of Ti–Al Laminated Composites"

_materials, 2021, doi:10.3390/ma14061323_

Round 1

Reviewer 1 Report

The manuscript entitled ‘Effects of SiC fiber and laminated structure on mechanical properties of Ti-Al laminated composites prepared by vacuum hot-pressing’ falls within the scope of the journal Materials.

In this manuscript, the effects of SiC fiber and laminated structure on the properties of Ti-Al laminated composites were studied. A novel method of fiber weaving was implemented to arrange the SiC fibers, which can guarantee the equal spacing of the fibers without introducing other elements. Results showed that with the higher exerted pressure, more compact structure with less Kirkendall holes can be obtained in SiCf-reinforced Ti/Ti-Al laminated composites

The paper contains very interesting experimental results as well as corresponding analyses. It is of sufficient scientific interest and has originality in its technical content to merit publication. The authors have cited the relevant literature. Methods, interpretations of results are correct. The authors presented extensive material supporting the conducted research. The issues were well presented. In terms of content, the analysis does not raise any objections. The arrangement of work maintains substantive continuity and constitutes a logical whole. The conclusions are supported by the results of the research. The conclusions are correct, new and they are supported by the results of the research.

However, the manuscript is not suitable for publication in its present form. This paper requires minor corrections (mainly editorial).

Comments and remarks are presented below.

  1. Table 1 should be completely on one page.
  2. Figures 2, 3, 5, 7, 8, 10, 12 - 18 should be enlarged.
  3. I suggest redrafting the first conclusion. In the first place, the effect of innovative technology should be presented. The sentence 'A novel method of fiber weaving was implemented to arrange the SiC fibers' is a description of the research, not a conclusion.

Reviewer 2 Report

The authors describe the fabrication and mechanical behavior of laminated fiber-reinforced Ti/Ti-Al composites. The laminates were fabricated from pure Al and Ti foils and Al foils with SiC fiber braids by hot pressing. Two different arrangements of the fiber braid Al foils were investigated. The results are novel and interesting, especially in situ tensile measurements. However, some comments should be addressed to the authors before the acceptance of manuscript.

Comments:

  1. Why did you use different pressures when sintering Ti-Al and SiC-reinforced Ti/Ti-Al composites? This makes it difficult to separate the effects of sintering pressure and the role of fibers under mechanical deformation.
  2. What PDF cards were used for XRD analysis? There are no reflections of Ti3Al phase at smaller angles (approx. 41 degree)? Or they overlap with Ti reflections? If there is a preferential orientation of crystallites, it is not typical for such far angles for Ti3Al.
  3. Line 163. Please change “Fig. 7a and 7b presents” to “Fig. 7 presents” or “Fig. 7a and 7b present”. Also, check other grammar errors.
  4. Fig. 9. The authors should check the results of the XRD. What was the X-ray penetration depth? Do you see peaks from the SiC fibers or it TiAl2 phase? What was the crystal structure of SiC fibers used (3C, 4H, 6H or other)?
  5. Table 2. EDX is semi-quantitative analysis. Please round your values or specify accuracy or error scales.
  6. Is the formation of microcracks near the fibers related to the volume expansion associated with the formation of TiC and Al4C3 phases? Please discuss it in the manuscript.
  7. The figures should have higher resolution since it hardly readable.

Reviewer 3 Report

The manuscript entitled “Effects of SiC fiber and laminated structure on mechanical properties of Ti-Al laminated composites prepared by vacuum hot-pressing” describes the mechanical properties of Ti/Ti-Al and SiC fiber reinforced Ti/Ti-Al laminated composites. The laminates were made using vacuum hot press. The effects of SiC fibers and laminated structure on the properties of Ti-Al laminated composites were studied. A novel method of fiber weaving was implemented to arrange the SiC fibers, without the use of different materials that could contaminate the laminate. Tensile and flexural testing was performed. The deformation behavior and fracture mechanisms of SiC fiber -reinforced Ti/Ti-Al laminated composites were obtained through in-situ tensile test. The tensile properties of the laminated composites could be effectively improved by introducing the SiC fibers. While the bending properties was not influenced probably due to the small volume fraction of fibers.

Overall, it is a well written and presented manuscript with some drawbacks.

Although the results are very well presented, the discussion needs some additional work.

Following are my suggestions to the authors:

  1. Title

I would change the title to : “Effects of SiC fibers and laminated structure on the mechanical properties of Ti-Al laminated composites”

  1. Abstract.

The abstract includes the sentences:

“With the growth and merging of micro-cracks, interlayer cracks formed. With the destruction of the outermost Ti layer, the composite started to fail.”

I do not believe that this information is necessary in the abstact.

Just the methodology and major conclusions are necessary.

Also, the last sentence about the low bending modulus should be omitted, as it makes no sense. Moreover,  you should not call it “bending modulus” but rather “flexural stiffness”

  1. Line 29 change to “modulus of Elasticity”

  1. Line 36 change “are extensively researched recently” to “have been extensively researched”

  1. Line 61 change “then hybridization with intermetallic layers” to “then hybridize with intermetallic layers”

  1. Line 74 change to the beginning of the last sentence to “the interface”

  1. At the end of the Introduction the authors should clearly mention the innovation of the presented study. Not just a synopsis of what they did but rather why? It is not really clear what the innovative aspects of this manuscript are.

  1. Line 16 change “di-rection” to “direction”

  1. Line 20 change “in-termetallic” to “intermetallic”

10 Line 92. Table 1. Should not break in two pages.

  1. Line 96. How were the holes drilled?

  1. Lines 95-100 The fibers were continuous right. In fig. 12 it seems that the fibers are short

  1. Figure 2. A cross section of the foil with the woven fibers would be helpful.

  1. Figure 3. The schematic of (a) is smaller compared to (b). It would look much better if both had the same size. Also, the Part 1 and Part 2 text should be larger. You should identify better the two parts.

  1. Figure 5 (c) shows a part 1 and part 2, but no explanation is provided. You should provide a comment regarding part 1 and 2.

  1. Line 157 you mention 10mm gauge length but in figure 6 you mention 15mm

  1. Line 165 change “among which scattered the SiC fibers (black dots).” To “among which are the scattered SiC fibers (black dots).”

  1. Lines 104-109 you should mention more clearly the difference between part 1 and part 2. It would be better to call it pattern A and pattern B instead of part.

  1. Line 180 change “previous researches” to “Previous studies”

  1. Line 188 change “This can be attributed to the larger exerted pressure (40 MPa) during the fabricating processes. While the pressure applied in the Ti/Ti-Al laminated composites was 10 MPa.” to “This may be attributed to the larger exerted pressure (40 MPa) during the fabricating processes, compared to the 10MPa pressure applied in the Ti/Ti-Al laminated composites.

  1. Line 194 “It means that “Ti-Al-SiCf-Al-Ti” laminated structure should be preferred to obtain the equal spacing of the fiber arrangement.” I do not understand your point. It seems to me that part 2 resulted in fiber aggregation that in turn resulted in more cracks around the fibers. Also, a zone (Al?) can be seen around the fibers, with significant radial cracking.

Thus part 2 looks much worse compared to part 1.

  1. How was the strain measured in uniaxial testing?

  1. Line 316 change to “fabricated laminated composites”

  1. Lines 297-305 I believe stage I should be omitted. A metallic material cannot exhibit an increase in flexural stiffness. The initial low stiffness makes no sense, it is a result of a non-perfect setup/specimen (and happens every single time). Essentially, the experiment begins in Stage 2. You should try to describe better what happens in Stages 2 to 5. Furthermore, the blue curve in Fig. 16 should be shifted left. Maybe you can add a table with the flexural stiffness of each material in each stage.

This discussion should be connected with the discussion in lines 316-324.

  1. Line 240 it is not a “stretch test” but a “uniaxial tension test”

  1. Lines 248-249 How do you come up with 1.2% ? Ultimate elongation is 1.6% vs 1.4%. Also you can find the yielding at 0.2% deformation.
